# Unlocking the Pre-Trained Model as a Dual-Alignment Calibrator for Post-Trained LLMs

## Abstract

Post-training boosts the performance of large language models (LLMs) but systematically degrades their confidence calibration, making them frequently over-confident. Recent post-hoc LLM calibration methods circumvent the challenge by aligning the post-trained language model with its pre-trained counterpart; however, they treat calibration as a static output distribution matching problem, and thus fail to capture the complex dynamics of post-training induced on calibration. Our investigation into these dynamics reveals that calibration errors stem from two distinct regimes: (i) *output drift*, where final confidence is inflated while intermediate decision process remains consistent, and (ii) *process drift*, where the intermediate pathways themselves diverge. Based on this diagnosis, we propose DUAL-ALIGN, a dynamic unsupervised framework performing dual alignment for LLM confidence calibration. It applies *output alignment* to correct output drift by matching the final output distributions. For process drift, it introduces novel *process alignment*, a technique that first identifies the specific layer where the models' inference paths diverge and then realigns the stability of their subsequent trajectories. This dual strategy enables learning a temperature parameter that corrects both calibration error types that occur during post-training. Experiment results demonstrate that our method brings consistent improvement compared with representative baselines, reducing calibration error and approaching the performance of a supervised oracle.

## 1 Introduction

Post-training methods such as instruction tuning and reinforcement learning from human feedback, substantially improves large language model (LLM) alignment and adaptability across tasks (Wei et al., 2022; Long Ouyang & et al., 2022; Zhang et al., 2025). Yet it also introduces new challenges in uncertainty estimates, often amplifying over-confidence relative to the pre-trained language models (PLMs) (Achiam et al., 2023; Shen et al., 2024). To circumvent this, researchers have explored confidence calibration, such as temperature scaling (TS) (Guo et al., 2017) for post-trained LMs (PoLMs): aligning predicted probabilities with empirical accuracy so models behave cautiously under uncertainty (Xiong et al., 2024).

Recent unsupervised methods, such as DACA (Luo et al., 2025a), use the PLM as a reference to calibrate the PoLM. To avoid potential conflicts from new knowledge introduced by post-training, DACA chooses to only align on samples where predictions are consistent between PLM and PoLM. However, this selective alignment strategy is inherently data-inefficient, as it discards all samples where the models disagree. More critically, by focusing solely on matching the final output confidence, it treats calibration as a static, surface-level matching problem. This fails to address the complex drifts in the model's intermediate inference process induced by post-training, which are often the root cause of miscalibration. We raise a key question here: *How does post-training alter the decision process of LLMs, and can we use that understanding to calibrate them more effectively?*

To answer this, we begin by investigating the different behavioral regimes of the PLM and PoLM by analyzing their layer-wise predictions and final outputs. Our analysis at Figure 1 reveals two distinct post-training phenomena: (i) In samples where the PoLM and PLM agree on the final answer, their intermediate decision process remains largely consistent, yet the PoLM's final confidence is systematically inflated—a phenomenon we term **output drift** (Figure 1(a)). (ii) Conversely, in samples where they disagree, the models' decision pathways diverge sharply at a specific intermediate

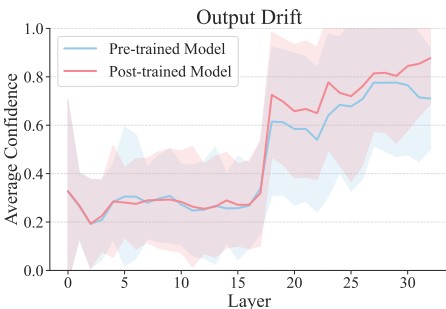 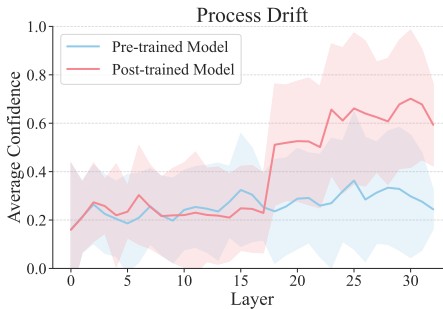

Figure 1: **Two post-training regimes underlying miscalibration.** (a) *Output drift*: the PLM and PoLM follow similar layer-wise trajectories, yet the PoLM's final confidence is inflated (agreement cases). (b) *Process drift*: the models' intermediate inference process diverges sharply from a specific layer, yielding different answers (disagreement cases). Curves are computed from layer-wise confidence trajectories projected by LogitLens (nostalgebraist, 2020) and are averaged over samples in MMLU (Hendrycks et al., 2021) (standard deviation shown in the shade region); see Section 3 for detailed illustration.

layer, causing their inference trajectories to split and lead to different answers. We term this more fundamental change **process drift** (Figure 1(b)). These observations motivate a calibration approach that addresses both phenomena at their source.

**Our contributions.** To this end, we propose DUAL-ALIGN, a dynamic post-hoc calibration framework (Figure 4) that treats calibration as a *dual alignment* problem. It performs (1) **output alignment** to correct surface-level overconfidence by matching the PoLM's final-layer output distribution with the PLM's. Our motivation for a deeper alignment stems from our key observation that post-training creates a problematic pattern where extreme overconfidence is coupled with unnaturally low Inferential Stability Entropy (ISE) (Figure 5) calculated over the LLM inference trajectory across different layers. To rectify this, our framework introduces a novel (2) **process alignment**, which first identifies the Peak Divergence Layer (PDL) where the models' inference pathways diverge, and then aligns the PoLM's ISE with the PLM's healthier distribution from that point onwards. Importantly, our framework interpolates between these two objectives on a per-sample basis using a divergence-derived weight, which yields a temperature parameter that adapts across different miscalibration regimes without labels. Empirically, we show that our method achieves substantial calibration improvements, reducing the Expected Calibration Error by over 30% across various LLM architectures compared to strong baselines.

## 2 PRELIMINARIES

**Confidence calibration for PoLMs.** We aim to calibrate a post-trained language model PoLM, denoted by $f$, using a pre-trained language model PLM, $g$, as a reference. In the context of a multiple-choice question, for a given input prompt $x$, the model produces final-layer logits $z_f^L(x)$ corresponding to the candidate choices. The model's prediction, $\hat{y}_f(x)$, is the choice with the highest probability derived from the logits via a softmax function, and this maximum probability value is taken as its confidence, $\hat{P}(x)$. A model is considered perfectly calibrated if its confidence matches its true accuracy, i.e., $\Pr\left(Y = \hat{y} \mid \hat{P} = \beta\right) = \beta$, where $Y$ is the ground-truth label.

A standard metric to measure this discrepancy is the Expected Calibration Error (ECE) (Naeini et al., 2015). In practice, ECE is estimated empirically by partitioning $K$ samples into $M$ bins $b_1, b_2, \ldots, b_M$ based on the model's predicted confidence scores, and then computed as:

$$\text{ECE} = \sum_{m=1}^{M} \frac{|b_m|}{K} \left| \text{acc}(b_m) - \text{conf}(b_m) \right|, \tag{1}$$

where $\text{acc}(b_m)$ and $\text{conf}(b_m)$ are the average accuracy and confidence in bin $b_m$. A smaller ECE indicates better calibration performance of the model. While PLMs are often well-calibrated, literature recognize that post-training often degrades this property, leading to overconfident predictions (Xiao et al., 2025; Luo et al., 2025a; Leng et al., 2025). Our experiments in Figure 2 verify this finding.

**Post-hoc calibration methods.** Post-hoc calibration adjusts a model's confidence without altering its predictions. A popular supervised method is Temperature Scaling (TS) (Guo et al., 2017), which

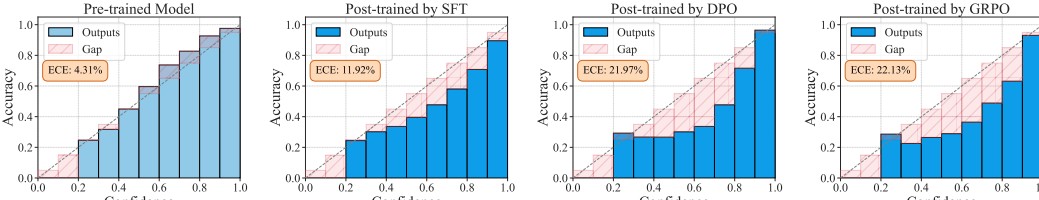

Figure 2: **Reliability diagrams on MMLU for a PLM vs. PoLMs obtained via different post-training methods.** The pre-trained model is Llama-3.1-8B and we consider Supervised Fine-tuning (SFT), Direct Preference Optimization (DPO) and Group Relative Policy Optimization (GRPO).

softens the probability distribution by applying a scalar temperature $\tau > 0$ to the final-layer logits:

$$p_f(y = j \mid \boldsymbol{x}, \tau) = \text{softmax}\left(\frac{\boldsymbol{z}_f^L(\boldsymbol{x})}{\tau}\right)_j. \tag{2}$$

The temperature $\tau$ is optimized on a labeled dataset. To eliminate the need for labels in calibration, unsupervised methods like DACA (Luo et al., 2025a) align the PoLM's confidence with that of the better-calibrated PLM. Crucially, DACA performs this alignment exclusively on samples where the models agree on the prediction, thereby avoiding under-confidence issues caused by optimizing on disagreement cases. However, it treats calibration as a static, surface-level matching problem. This fails to address the complex drifts in the model's intermediate inference process induced by post-training, which is the focus of our paper.

## 3 UNDERSTANDING THE EFFECTS OF POST-TRAINING ON CALIBRATION

In this section, we aim to understand how post-training affects the calibration performance of LLMs based on their internal inference processes. Let the input prompt be a sequence of tokens $\boldsymbol{x} = \{x_1, x_2, \dots, x_N\}$. Our analysis focuses on the final token, $x_N$, as its hidden state is used to generate the model's prediction. At each layer $l \in [1, L]$ of a transformer model (Vaswani et al., 2017), the hidden state for this token is conceptually updated as:

$$\boldsymbol{h}^l(x_N) = \boldsymbol{h}^{l-1}(x_N) + \text{Attn}^l(x_N) + \text{MLP}^l(x_N), \tag{3}$$

where $\boldsymbol{h}^l \in \mathbb{R}^{d_{\text{model}}}$ denotes the hidden state at the $l$-th layer. Using LogitLens (nostalgebraist, 2020), we can project any intermediate hidden state $\boldsymbol{h}^l(x_N)$ into the vocabulary space via the unembedding matrix $W_U \in \mathbb{R}^{V \times d_{\text{model}}}$, with $V$ as the vocabulary size. Since the embedding $\boldsymbol{h}^l(x_N)$ encapsulates information from the entire input $\boldsymbol{x}$, we denote the resulting per-layer logits as $\boldsymbol{z}^l(\boldsymbol{x}) = W_U \boldsymbol{h}^l(x_N) \in \mathbb{R}^V$, from which we can derive a probability distribution $\boldsymbol{p}^l(\boldsymbol{x})$ at every layer by applying softmax.

To understand how post-training alters an LLM's decision process, we analyze the layer-wise information of a pre-trained model $g$ and its post-trained counterpart $f$. Our method involves two components: we first track the evolution of predictive confidence across layers, and second, to symmetrically measure the predictive distance between the models at each layer, we use the Jensen-Shannon Divergence (JSD), denoted as $d^l(\boldsymbol{x}) = D_{JS}(\boldsymbol{p}_g^l(\boldsymbol{x}) \,\|\, \boldsymbol{p}_f^l(\boldsymbol{x}))$. This dual analysis, when performed separately on samples grouped by whether the models' final predictions agree or disagree, reveals two distinct post-training effects on model calibration:

**Output drift.** Occurring predominantly on agreed samples, output drift describes the scenario where the PoLM's intermediate decision process remains consistent with the PLM. As shown in Figure 1 (a), their confidence trajectories follow a similar path where confidence sharply increases in later layers, although the PoLM is systematically

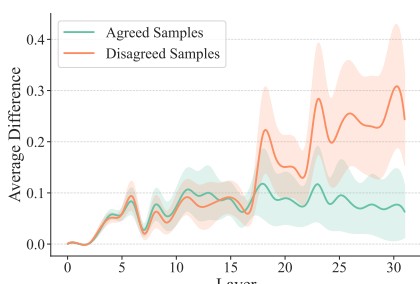

Figure 3: **Layer-wise predictive distance between PLM and PoLM.** We plot the predictive distance ($d^l(\boldsymbol{x})$) between $\boldsymbol{p}_g^l$ and $\boldsymbol{p}_f^l$. Agreement samples show low difference while disagreement samples exhibit a sharp spike at an intermediate layer, indicating process drift.

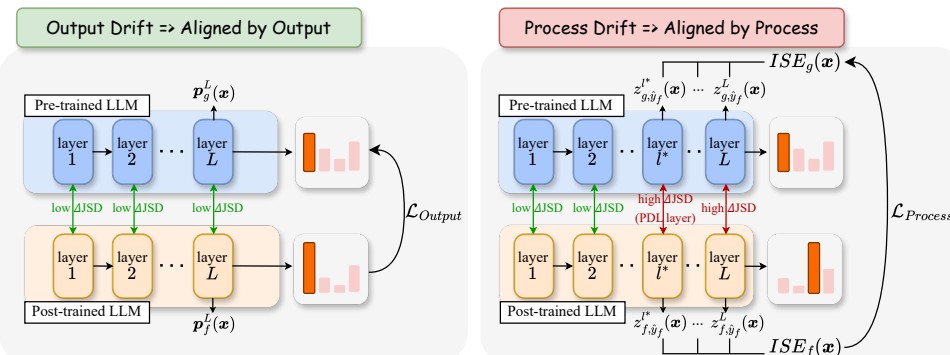

Figure 4: **Illustration of our method DUAL-ALIGN**. Our approach takes care of both output drift by focusing strategically on aligning the LLMs' output confidence with $\mathcal{L}_{\text{Output}}$ (Left), and process drift by firstly identifying the Peak Divergence Layer (PDL) and then aligning the Inferential Stability Entropy (ISE) calculated w.r.t. the process drift between PLM and PoLM with the learning objective $\mathcal{L}_{\text{Process}}$ (Right).

overconfident in the final outputs. This phenomenon is further confirmed by the consistently low JSD between their intermediate logit distributions projected by the unembedding matrix, as shown in Figure 3. In this regime, post-training has primarily altered the final output distribution rather than the inference pathway.

**Process drift.** A more fundamental drift *that is overlooked in literature*, termed process drift, is usually observed on disagreed samples, where the PoLM's layer-wise inference process diverges sharply from PLM. A critical feature, visible in Figure 3, is that the predictive distance $d^l(\boldsymbol{x})$ between PoLM and PLM is low in the early layers but then exhibits an obvious increase at an intermediate layer, which might signal an abrupt difference in inferential strategy. This divergence is also evident in the confidence trajectories shown in Figure 1(b), where the two models' layer-wise confidence scores are closely aligned in early layers, but then split apart at an intermediate stage. Our analysis thus suggests that naively aligning the final outputs of PLM and PoLM on all disagreement samples would be counterproductive, as it forces a match between outputs generated from fundamentally different intermediate decision processes, which can ultimately harm calibration.

## 4 PROPOSED FRAMEWORK: DUAL-ALIGN

Our analysis in Section 3 reveals that post-training induces two distinct phenomena: output drift, where output confidence becomes inflated in PoLM while the the intermediate computations remain similar to PLM, and process drift, where the model's inference pathway fundamentally diverges. Motivated by these findings, we propose DUAL-ALIGN (Figure 4), a novel post-hoc LLM calibration framework designed to address both effects in a synergistic manner. Our approach aims to learn a temperature parameter $\tau$ that effectively calibrates the post-trained model by comprehensively accounting for these underlying drifts, using only unlabeled data.

### 4.1 OUTPUT ALIGNMENT FOR OUTPUT DRIFT

When post-training primarily causes a output drift, the PoLM and PLM arrive at the same answer, but the PoLM exhibits inflated confidence in its output. In these circumstances, the PLM's final-layer output distribution serves as a reliable and well-calibrated target. We address this with a **output alignment** objective, which aims to correct the PoLM's overconfidence directly. This is achieved by minimizing the KL divergence between the temperature-scaled final-layer output distribution of the PoLM ($f$) and the original distribution of the PLM ($g$):

$$\mathcal{L}_{\text{Output}}(\tau; \boldsymbol{x}) = D_{KL}(\boldsymbol{p}_g^L(\boldsymbol{x}) \,||\, \boldsymbol{p}_f^L(\boldsymbol{x}, \tau)). \tag{4}$$

As depicted in the left panel of Figure 4, this loss component encourages the PoLM's temperature-scaled confidence scores to mirror those of the better-calibrated PLM, effectively correcting the output confidence miscalibration introduced during post-training.

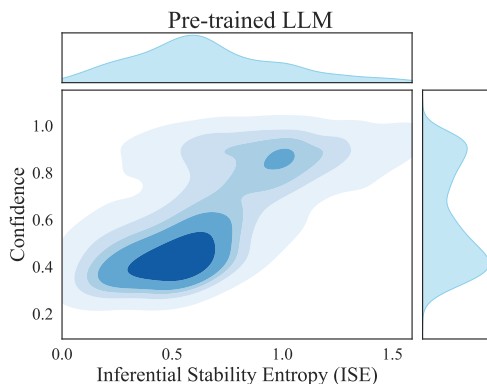 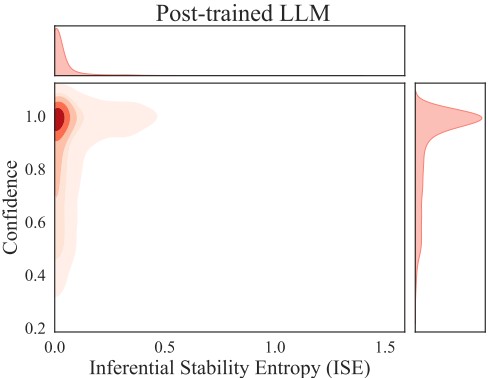

Figure 5: **Relationship between output confidence and Inferential Stability Entropy (ISE).** The pre-trained model (left) shows healthy uncertainty distribution, while the post-trained model (right) exhibits extreme overconfidence coupled with unnaturally low ISE values, indicating rigid conviction processes.

### 4.2 PROCESS ALIGNMENT FOR PROCESS DRIFT

A process drift represents a more significant alteration, where the PoLM's intermediate decision process diverges sharply from the PLM's, resulting in a different final answer. For such cases, naively enforcing output alignment is counterproductive; as aligning the final output or even the LLM representations between PoLM and PLM would force the PoLM to match a conclusion derived from a fundamentally different inference process, leading to severe underconfidence. Instead, our key insight is to regularize the PoLM's intermediate inference process itself. Specifically, we propose to align the *stability* of the model inference that occurs after the point of divergence. This ensures that even when the PoLM reaches a different conclusion, its conviction in that conclusion emulates the properly stable confidence characteristic of the well-calibrated PLM, preventing erratic overconfidence.

To implement this, we first identify the exact layer where the two models' inference pathways diverge most sharply by first measuring their per-layer output distance using the JSD. We then define the Peak Divergence Layer (PDL), $l^*(\boldsymbol{x})$, as the layer exhibiting the maximum *increase* in JSD from the previous one:

$$l^*(\boldsymbol{x}) = \underset{l \in \{2,\ldots,L\}}{\arg\max} \left( D_{JS}(\boldsymbol{p}_f^l(\boldsymbol{x}) \,||\, \boldsymbol{p}_g^l(\boldsymbol{x})) - D_{JS}(\boldsymbol{p}_f^{l-1}(\boldsymbol{x}) \,||\, \boldsymbol{p}_g^{l-1}(\boldsymbol{x})) \right). \quad (5)$$

The measurement of a model's conviction stability begins by identifying the final prediction of the post-trained model, $\hat{y}_f(\boldsymbol{x})$, and the Peak Divergence Layer ($l^*$). For each layer $l$ from $l^*$ to the final layer $L$, the logit vector from the post-trained model, $\boldsymbol{z}_f^l(\boldsymbol{x})$, is generated. From this vector, the specific logit value corresponding to the position of the final prediction is extracted, which is denoted as $z_{f,\hat{y}_f}^l(\boldsymbol{x})$. These individual logit values are then collected to form a vector, $\boldsymbol{v}_f(\boldsymbol{x}) = [z_{f,\hat{y}_f}^{l^*}(\boldsymbol{x}), z_{f,\hat{y}_f}^{l^*+1}(\boldsymbol{x}), \ldots, z_{f,\hat{y}_f}^L(\boldsymbol{x})]$. After normalizing with softmax, The stability is then quantified by calculating an entropy value from this sequence of logits with the following formula:

$$\text{ISE}_f(\boldsymbol{x}) = -\sum_{l=l^*}^{L} q_f^l(\boldsymbol{x}) \log q_f^l(\boldsymbol{x}), \qquad q_f^l(\boldsymbol{x}) = \frac{\exp\left(v_f^l(\boldsymbol{x})\right)}{\sum_{j=l^*}^{L} \exp\left(v_f^j(\boldsymbol{x})\right)}, \qquad l = l^*, \ldots, L. \quad (6)$$

Our motivation for this approach is rooted in the hypothesis that a PoLM's overconfidence stems from its conviction process becoming overly rigid, where it quickly settles on a decision with consistently high confidence, unlike the more deliberative PLM. A lower ISE signifies a more consistent conviction across intermediate layers, and this hypothesis is supported by the empirical observations in Figure 5.

We first observe that the PLM's output confidence is distributed across a reasonable range, reflecting a healthy level of uncertainty (Left). In sharp contrast, the PoLM suffers from severe overconfidence, with its confidence scores overwhelmingly concentrated near 1.0 (Right). Furthermore, the two models show a vastly different relationship between confidence and inferential stability. For the PLM, confidence is largely stable across its typical ISE range. The PoLM, however, exhibits an undesirable correlation where extreme confidence is systematically coupled with unnaturally low ISE.

This suggests the PoLM's conviction process has become over-certain and with less variations across different layers, which is reflected in Figure 5 by the dense clustering of data points in the top-left corner of the plot, where confidence approaches 1.0 as ISE nears 0.

This sharp contrast between PoLM and PLM reveals that simply correcting the final output confidence may be insufficient. A better approach is to address the intermediate inference dynamics, which makes the PLM's healthier ISE distribution an ideal target. Our process alignment loss is therefore designed to restore a more stable conviction process for PoLM by minimizing the squared difference between the ISE of the two models:

$$\mathcal{L}_{\text{Process}}(\tau; \boldsymbol{x}) = \left(\text{ISE}_f(\boldsymbol{x}, \tau) - \text{ISE}_g(\boldsymbol{x})\right)^2, \tag{7}$$

where we divide the PoLM logits by a temperature $\tau$ to calculate $\text{ISE}_f(\boldsymbol{x}, \tau)$. This objective optimizes $\tau$ to align the stability of the PoLM's inference process with that of a better-calibrated PLM.

### 4.3 DUAL-ALIGN: A UNIFIED CALIBRATION FRAMEWORK

DUAL-ALIGN addresses the two miscalibration errors incurred by LLM post-training in one unified manner. Specifically, we achieve this by using the magnitude of the peak JSD increase, $\Delta D_{JS}^{l^*}(\boldsymbol{x}) = D_{JS}(p_f^{l^*}(\boldsymbol{x}) \,\|\, p_g^{l^*}(\boldsymbol{x})) - D_{JS}(p_f^{l^*-1}(\boldsymbol{x}) \,\|\, p_g^{l^*-1}(\boldsymbol{x}))$, as a natural indicator of the process drift's severity for each sample. The final learning objective is a weighted combination of the output and process alignment components:

$$\mathcal{L}_{\text{DUAL-ALIGN}}(\tau; \boldsymbol{x}) = (1 - \Delta D_{JS}^{l^*}(\boldsymbol{x})) \cdot \mathcal{L}_{\text{Output}}(\tau; \boldsymbol{x}) + \Delta D_{JS}^{l^*}(\boldsymbol{x}) \cdot \mathcal{L}_{\text{Process}}(\tau; \boldsymbol{x}). \tag{8}$$

This unified objective [1] uses the model's intermediate predictive divergence $\Delta D_{JS}^{l^*}(\boldsymbol{x})$ as a data-driven weight coefficient during training. In this way, the loss function dynamically balances the two alignment objectives for each sample, without introducing separate hyperparameter. By minimizing the expected loss $\mathbb{E}_{\boldsymbol{x} \in \mathcal{D}}[\mathcal{L}_{\text{DUAL-ALIGN}}(\tau; \boldsymbol{x})]$ over an unlabeled dataset $\mathcal{D} = \{\boldsymbol{x}_i\}_{i=1}^K$, DUAL-ALIGN learns an optimal temperature $\tau^*$ that can comprehensively handle the post-training effects on LLM calibration. During inference, we apply the learned $\tau^*$ to calibrate PoLMs in their final outputs, which does not require additional computational cost or PLMs.

## 5 EXPERIMENTS

In this section, we present empirical evidence to validate the effectiveness of our method across various LLM architectures and datasets. We describe the setup in Section 5.1, followed by the results and comprehensive analyses in Section 5.2–Section 6.

### 5.1 EXPERIMENTAL SETUP

**Models, datasets and evaluation.** Our evaluation comprehensively assesses a diverse array of large language models, encompassing various scales and architectures, including the Llama-3.1 series (Grattafiori et al., 2024), the Gemma-3 series (Team et al., 2025) and the Qwen-2.5 series (Yang et al., 2024a). More details about these LLMs are presented in Appendix A.1.

We validate our methodology's efficacy across three widely-adopted evaluation benchmarks: MMLU (Hendrycks et al., 2021), and MedMCQA (Pal et al., 2022). All benchmark datasets are obtained from the Hugging Face repository. Comprehensive descriptions of each evaluation dataset are provided in Appendix A.2.

To assess the calibration performance of DUAL-ALIGN, we measure four established metrics: Expected Calibration Error (**ECE**)(Naeini et al., 2015), Maximum Calibration Error (**MCE**) (Naeini et al., 2015), Adaptive Calibration Error (**ACE**) (Nixon et al., 2019) and **Brier Score** (Brier, 1950). Additional evaluation details are provided in Appendix A.3.

**Baselines.** We compare our method with several post-hoc calibration techniques. Our unsupervised baselines include **DACA** (Luo et al., 2025a), which aligns the pre-trained model on agreement samples; a hidden-state-based approach, Internal Consistency (**IC**) (Xie et al., 2024b), which measures the ratio of consistency between each layer's predictions and the final layer's output; and two prompt-based methods: **CAPE** (Jiang et al., 2023), which reduces bias by reordering answer choices, and

---

[1] We adopt base-2 logs in JSD calculation to ensure its $\Delta D_{JS} \leq 1$.

| Models | Methods | Evaluation Metrics | | | |
|--------|---------|---------------|---------------|---------------|---------------|
| | | ECE (%) ↓ | MCE (%) ↓ | ACE (%) ↓ | Brier Score ↓ |
| Llama3.1-8B | Vanilla | $10.806_{\pm 0.275}$ | $18.602_{\pm 0.212}$ | $11.809_{\pm 0.652}$ | $0.461_{\pm 0.005}$ |
| | CAPE | $12.567_{\pm 0.134}$ | $20.788_{\pm 0.841}$ | $13.134_{\pm 0.257}$ | $0.495_{\pm 0.001}$ |
| | Elicitation | $13.203_{\pm 0.067}$ | $40.983_{\pm 4.065}$ | $21.300_{\pm 1.714}$ | - |
| | IC | $11.716_{\pm 0.248}$ | $64.448_{\pm 29.949}$ | $19.517_{\pm 3.165}$ | - |
| | DACA | $7.811_{\pm 0.619}$ | $13.824_{\pm 0.667}$ | $8.064_{\pm 0.544}$ | $0.451_{\pm 0.004}$ |
| | **DUAL-ALIGN (Ours)** | $\mathbf{2.871_{\pm 0.308}}$ | $\mathbf{5.587_{\pm 0.648}}$ | $\mathbf{3.222_{\pm 0.306}}$ | $\mathbf{0.445_{\pm 0.004}}$ |
| | TS† (oracle) | $1.526_{\pm 0.450}$ | $4.790_{\pm 1.090}$ | $1.985_{\pm 0.609}$ | $0.441_{\pm 0.004}$ |
| Qwen2.5-14B | Vanilla | $16.735_{\pm 0.375}$ | $32.406_{\pm 0.583}$ | $21.848_{\pm 1.130}$ | $0.388_{\pm 0.006}$ |
| | CAPE | $18.022_{\pm 0.061}$ | $36.091_{\pm 0.501}$ | $20.987_{\pm 0.340}$ | $0.407_{\pm 0.001}$ |
| | Elicitation | $15.321_{\pm 0.002}$ | $85.556_{\pm 0.000}$ | $31.973_{\pm 2.713}$ | - |
| | IC | $32.852_{\pm 0.258}$ | $47.360_{\pm 5.4265}$ | $22.089_{\pm 0.627}$ | - |
| | DACA | $5.146_{\pm 0.340}$ | $\mathbf{8.867_{\pm 0.590}}$ | $4.427_{\pm 0.287}$ | $0.329_{\pm 0.004}$ |
| | **DUAL-ALIGN (Ours)** | $\mathbf{2.423_{\pm 0.070}}$ | $11.241_{\pm 2.918}$ | $\mathbf{3.602_{\pm 0.642}}$ | $\mathbf{0.326_{\pm 0.005}}$ |
| | TS† (oracle) | $2.297_{\pm 0.124}$ | $11.411_{\pm 2.996}$ | $3.986_{\pm 0.994}$ | $0.326_{\pm 0.005}$ |
| Gemma-3-27B | Vanilla | $23.842_{\pm 0.336}$ | $58.230_{\pm 8.103}$ | $35.240_{\pm 2.461}$ | $0.481_{\pm 0.007}$ |
| | CAPE | $19.891_{\pm 0.053}$ | $38.791_{\pm 0.334}$ | $23.281_{\pm 0.345}$ | $0.445_{\pm 0.01}$ |
| | Elicitation | $18.413_{\pm 0.284}$ | $26.526_{\pm 2.564}$ | $22.456_{\pm 1.326}$ | - |
| | IC | $36.667_{\pm 0.313}$ | $53.937_{\pm 0.414}$ | $36.746_{\pm 0.346}$ | - |
| | DACA | $16.842_{\pm 0.324}$ | $35.205_{\pm 0.660}$ | $23.985_{\pm 0.524}$ | $0.406_{\pm 0.006}$ |
| | **DUAL-ALIGN (Ours)** | $\mathbf{5.247_{\pm 0.310}}$ | $\mathbf{18.065_{\pm 8.913}}$ | $\mathbf{9.175_{\pm 1.565}}$ | $\mathbf{0.379_{\pm 0.005}}$ |
| | TS† (oracle) | $5.225_{\pm 0.254}$ | $18.069_{\pm 9.148}$ | $8.871_{\pm 1.561}$ | $0.359_{\pm 0.005}$ |

Table 1: **Main evaluation results on MMLU datasets across different LLMs**. Lower values indicate better performance. Best results among unsupervised methods are shown in **bold**. "IC": Internal-consistency; "TS": Temperature Scaling. † indicates calibration methods with access to labels. Values are percentages averaged over 3 runs.

**Elicitation** (Tian et al., 2023), which prompts the model to state its confidence. We also report results for the uncalibrated **Vanilla** model and use supervised **Temperature Scaling (TS)** (Guo et al., 2017) as an oracle. More details of baselines are presented in Appendix A.4.

## 5.2 MAIN RESULTS

**DUAL-ALIGN consistently achieves state-of-the-art results.** DUAL-ALIGN demonstrates superior performance across all evaluated models and metrics, establishing a new state-of-the-art for unsupervised LLM calibration by outperforming all other unsupervised baselines, as shown in Table 1. For instance, on MMLU with the Llama-3.1-8B, our method achieves an ECE of just 2.871%, a significant reduction compared to the 7.811% of the strongest unsupervised baseline, DACA, and the 10.806% of the uncalibrated model. Notably, our framework's performance can significantly outperform the hidden-state-based approach IC and closely approach that of the supervised TS oracle. This indicates that our method that tackles both output drift and process drift in a dual alignment manner, can effectively address the complex dynamics of miscalibration while reducing human annotation costs. We also present the reliability diagrams visualization in Appendix D.

**DUAL-ALIGN is effective across different model architectures and sizes.** To validate the scalability and generalizability of our method, we conduct experiments across different model architectures (Qwen2.5-14B and Gemma-3-27B) in Table 1, and the Qwen-2.5 model series with varying sizes in Table 2. The results demonstrate that our method can maintain its effectiveness as model architecture varies and model size increases from 7B to 32B parameters. In all configurations, our method consistently outperforms both the uncalibrated model and the DACA baseline. This consistent performance advantage across

| Size | Method | ECE (↓) | MCE (↓) |
|------|--------|---------|---------|
| 7B | Vanilla | $20.666_{\pm 0.382}$ | $38.647_{\pm 1.219}$ |
| | DACA | $10.312_{\pm 0.502}$ | $16.884_{\pm 0.954}$ |
| | DUAL-ALIGN | $\mathbf{9.406_{\pm 0.577}}$ | $\mathbf{15.256_{\pm 0.993}}$ |
| 14B | Vanilla | $23.842_{\pm 0.336}$ | $58.230_{\pm 8.103}$ |
| | DACA | $5.146_{\pm 0.340}$ | $\mathbf{8.867_{\pm 0.590}}$ |
| | DUAL-ALIGN | $\mathbf{2.423_{\pm 0.070}}$ | $11.241_{\pm 2.918}$ |
| 32B | Vanilla | $11.338_{\pm 0.065}$ | $23.522_{\pm 5.214}$ |
| | DACA | $10.958_{\pm 0.670}$ | $17.312_{\pm 1.082}$ |
| | DUAL-ALIGN | $\mathbf{9.203_{\pm 0.055}}$ | $\mathbf{15.723_{\pm 0.332}}$ |

Table 2: **Evaluation of DUAL-ALIGN with different model sizes.** We experiment with Qwen2.5 series of different model sizes.

different model scenarios highlights that DUAL-ALIGN is not tailored to a specific model but is a general solution that can be applied practically and flexibly.

## 5.3 ABLATION STUDY

To validate the key components of our DUAL-ALIGN framework, we conduct a series of ablation studies on the MMLU benchmark using the Llama-3.1-8B model. We investigate the contributions of our dual-component loss function and our dynamic layer selection strategy.

**Ablation on loss components.** To validate our dual-component loss, we compare the full DUAL-ALIGN framework against versions using only the output alignment loss ($\mathcal{L}_{\text{Output}}$) or the process alignment loss ($\mathcal{L}_{\text{Process}}$). As shown in Table 3, the "Output Only" variant is ineffective, performing worse than the DACA baseline. While the "Process Only" variant substantially reduces calibration error, our full DUAL-ALIGN framework—which dynamically integrates both losses—achieves the best performance. It significantly outperforms both ablated versions and approaches the supervised TS oracle, confirming the necessity of our dual-component strategy for effective calibration.

| Method | ECE (%) ↓ | MCE (%) ↓ | ACE (%) ↓ | Brier Score ↓ |
|---|---|---|---|---|
| Vanilla | $10.806_{\pm 0.275}$ | $18.602_{\pm 0.212}$ | $11.809_{\pm 0.652}$ | $0.461_{\pm 0.005}$ |
| DACA | $7.811_{\pm 0.619}$ | $13.824_{\pm 0.667}$ | $8.064_{\pm 0.544}$ | $0.451_{\pm 0.004}$ |
| DUAL-ALIGN (Output Only) | $10.267_{\pm 0.925}$ | $17.599_{\pm 1.145}$ | $10.393_{\pm 0.795}$ | $0.459_{\pm 0.003}$ |
| DUAL-ALIGN (Process Only) | $6.082_{\pm 1.982}$ | $9.082_{\pm 3.011}$ | $6.092_{\pm 1.925}$ | $0.449_{\pm 0.006}$ |
| **DUAL-ALIGN (Ours)** | $\mathbf{2.871_{\pm 0.308}}$ | $\mathbf{5.587_{\pm 0.648}}$ | $\mathbf{3.222_{\pm 0.306}}$ | $\mathbf{0.445_{\pm 0.004}}$ |
| TS$^{\dagger}$ (Oracle) | $1.526_{\pm 0.450}$ | $4.790_{\pm 1.090}$ | $1.985_{\pm 0.609}$ | $0.441_{\pm 0.004}$ |

Table 3: **Ablation study on the loss components of DUAL-ALIGN using Llama-3.1-8B on the MMLU datasets.** Our full, dual alignment method significantly outperforms the ablated versions, highlighting the necessity of addressing both output and process drift.

**Ablation on layer selection.** To validate our dynamic Peak Divergence Layer (PDL) selection strategy, we compare it against starting process alignment at fixed network depths ($L/4$, $L/2$, and $3L/4$). As shown in Table 4, our dynamic approach, which identifies the layer with the maximum JSD increase, yields substantially better calibration performance than any fixed-layer strategy. This result confirms that divergence is sample-dependent and that accurately identifying this layer on a per-sample basis is critical to the success of the DUAL-ALIGN framework.

| Method | ECE (%) ↓ | MCE (%) ↓ | ACE (%) ↓ | Brier Score ↓ |
|---|---|---|---|---|
| Vanilla | $10.806_{\pm 0.275}$ | $18.602_{\pm 0.212}$ | $11.809_{\pm 0.652}$ | $0.461_{\pm 0.005}$ |
| DACA | $7.811_{\pm 0.619}$ | $13.824_{\pm 0.667}$ | $8.064_{\pm 0.544}$ | $0.451_{\pm 0.004}$ |
| DUAL-ALIGN ($L/4$) | $4.716_{\pm 0.397}$ | $9.089_{\pm 1.298}$ | $5.087_{\pm 0.317}$ | $0.449_{\pm 0.004}$ |
| DUAL-ALIGN ($L/2$) | $4.862_{\pm 0.363}$ | $9.235_{\pm 0.874}$ | $5.228_{\pm 0.360}$ | $0.449_{\pm 0.003}$ |
| DUAL-ALIGN ($3L/4$) | $2.846_{\pm 0.460}$ | $5.806_{\pm 0.845}$ | $3.125_{\pm 0.587}$ | $0.446_{\pm 0.004}$ |
| **DUAL-ALIGN (Ours)** | $\mathbf{2.382_{\pm 0.619}}$ | $\mathbf{4.928_{\pm 1.030}}$ | $\mathbf{2.697_{\pm 0.715}}$ | $\mathbf{0.445_{\pm 0.004}}$ |
| TS$^{\dagger}$ (Oracle) | $1.526_{\pm 0.450}$ | $4.790_{\pm 1.090}$ | $1.985_{\pm 0.609}$ | $0.441_{\pm 0.004}$ |

Table 4: **Ablation study on the PDL selection strategy of DUAL-ALIGN using Llama-3.1-8B on the MMLU datasets.** Our proposed method, which selects the layer with the maximum JSD increase, yields the best calibration performance.

## 6 DISCUSSIONS

In this section, we explore the broader applicability and potential extensions of our proposed DUAL-ALIGN framework. We demonstrate its adaptability by showing its effectiveness on open-ended generation tasks, its successful generalization to specialized domains like medicine (see Appendix B for full results), and its compatibility with various post-training methodologies.

**Can DUAL-ALIGN be used for open-ended tasks?** While DUAL-ALIGN is designed for multiple-choice questions, it extends to open-ended tasks through reformulation. We convert open-ended generation into binary classification: the model first generates a free-form answer, then evaluates it via self-assessment. This approach follows the $p(\text{true})$ framework (Kadavath et al., 2022), effectively repurposing open-ended outputs for calibration without modifying our core method. We use

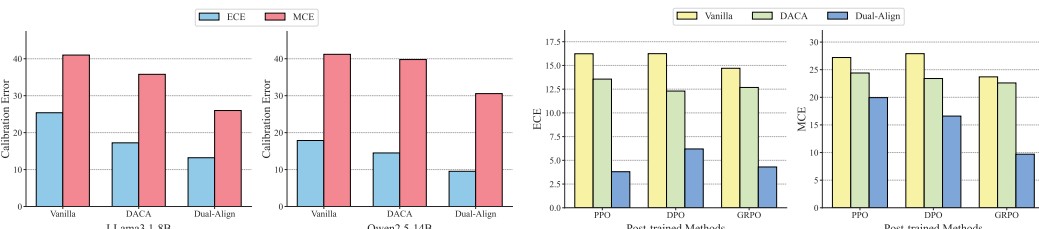

(a) **Applicability to open-ended question answering.** We evaluate LLama3.1 and Qwen2.5-14B on Truth-fulQA dataset.

(b) **Applicability to different post-training methods.** Apart from instruction-tuning, we consider PPO, DPO and GRPO training on Qwen2.5-7B.

TruthfulQA (Lin et al., 2022b). As shown in Figure 6a, DUAL-ALIGN significantly reduces both ECE and MCE on the TruthfulQA dataset for both LLama-3.1-8B and Qwen2.5-14B models. This demonstrates that our framework successfully adapts to open-ended generation, outperforming the strong DACA baseline and proving its versatility beyond multiple-choice formats.

**Applicability to other post-training methods.** To demonstrate the general applicability of our DUAL-ALIGN framework, we evaluate its performance on models subjected to various popular post-training techniques. We test on Qwen2.5-7B model trained with Proximal Policy Optimization(PPO) (Schulman et al., 2017), Direct Preference Optimization (DPO) (Rafailov et al., 2023), and Group Relative Policy Optimization (GRPO) (Liu et al., 2024a). As shown in Figure 6b, DUAL-ALIGN consistently outperforms both the uncalibrated model and the DACA baseline across all three methods. This robust performance highlights that our approach is not confined to a single post-training paradigm like instruction-tuning but generalizes effectively to models refined through various LLM post-training techniques, confirming its broad applicability.

## 7 RELATED WORKS

**Post-training** refines LLMs after their initial pre-training on broad datasets (Tie et al., 2025; Kumar et al., 2025). This stage includes methods like full fine-tuning for task-specific adaptation (Yue et al., 2023; Luo et al., 2025b), Parameter-Efficient Fine-Tuning (PEFT) such as LoRA for resource-efficient specialization (Hu et al., 2022; Gao et al., 2023; Trung et al., 2024), and reinforcement learning techniques like RLHF and DPO to align models with user preferences (Long Ouyang & et al., 2022; Rafailov et al., 2023). While creating versatile and aligned models, these post-training processes can introduce miscalibration. Our paper therefore investigates these effects and proposes a novel framework to calibrate Post-trained Language Models.

**Confidence calibration** aims to ensure a model's output confidence accurately reflects its correctness likelihood (Guo et al., 2017). However, studies show that post-training often leads to overconfident LLMs (Xiao et al., 2022; Chen et al., 2023; Liu et al., 2024b; Jiang et al., 2023). Current calibration approaches include eliciting verbalized confidence through prompting or fine-tuning (Lin et al., 2022a; Tian et al., 2023; Yang et al., 2024b; Xie et al., 2024a; Leng et al., 2025; Damani et al., 2025; Tao et al., 2025), and estimating confidence from output logits (Shen et al., 2024; Luo et al., 2025a; Vejendla et al., 2025). Closest to our work, Shen et al. (2024); Xie et al. (2024a) leverage hidden states for calibration. However, they fail to account for both the output / process drifts and alignment dynamics induced by post-training in one unified framework, which are central to our research.

## 8 CONCLUSION

In this paper, we tackle the overconfidence issue in post-trained LLMs, diagnosing that miscalibration stems from two distinct phenomena: output drift and process drift. We propose DUAL-ALIGN, an unsupervised post-hoc framework that performs a dual alignment to address both issues. The framework corrects output drift by matching final output distributions and rectifies process drift by identifying a Peak Divergence Layer and aligning the subsequent Inferential Stability Entropy. Critically, DUAL-ALIGN dynamically weighs these two objectives based on the model's intermediate predictive divergence, learning a single temperature parameter without human annotation. Experiments show our method achieves the state-of-the-art performance across diverse LLM architectures and datasets. We hope our work will inspire future research on understanding the LLM post-training effects on model calibration.

## REPRODUCIBILITY STATEMENT

We summarize our efforts below to facilitate reproducible results:

1. **Datasets.** We use publicly available datasets, which are described in detail in Section 5.1, and Appendix A.2.

2. **Baselines.** The description and hyperparameters of the LLM calibration baselines are explained in Appendix A.3, and Appendix A.4.

3. **Methodology.** Our method is fully documented in Section 4. Hyperparameters are specified in Appendix A.3.

4. **Open source.** Code, datasets and model checkpoints will be made publicly available for reproducible research.

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

# Appendix

## A  EXPERIMENTAL DETAILS

### A.1  MODELS DETAILS

We conduct our experiments across a diverse set of large language models, spanning various architectures and scales from prominent model families. Table 5 provides a detailed overview of the specific pre-trained and post-trained versions used in this study.

| Model Family | Model Type | HuggingFace Path |
|---|---|---|
| Llama-3.1 Family | Pre-trained Model | `meta-llama/Llama-3.1-8B` |
|  | Post-trained Model | `meta-llama/Llama-3.1-8B-Instruct` |
| Qwen-2.5 Family | Pre-trained Model | `Qwen/Qwen2.5-14B` |
|  | Post-trained Model | `Qwen/Qwen2.5-14B-Instruct` |
| Gemma-3 Family | Pre-trained Model | `google/gemma-3-27b-pt` |
|  | Post-trained Model | `google/gemma-3-27b-it` |

Table 5: An overview of models used in our experiments, detailing the pre-trained and post-trained versions and their respective Hugging Face paths for each family.

### A.2  DATASETS DETAILS

We evaluate our method on three diverse benchmarks. MMLU (Hendrycks et al., 2021) is a widely-adopted benchmark for measuring massive multitask language understanding. MedMCQA (Pal et al., 2022) is a large-scale, multi-subject, multiple-choice question dataset designed for the medical domain. TruthfulQA (Lin et al., 2022b) is a benchmark used to measure a model's truthfulness and its ability to avoid generating falsehoods.

For all datasets, we divide the data into a 30% subset for alignment training and a 70% test set. All three datasets are publicly available on Hugging Face[2]. For MMLU, we use the test split from all subjects, while for MedMCQA, we use the validation split.

### A.3  IMPLEMENTATION DETAILS

All results are reported as mean $\pm$ standard deviation from three independent runs with different random seeds. All post-hoc methods requiring optimization—including our supervised oracle (Temperature Scaling) and the unsupervised baselines (DACA, DUAL-ALIGN)—are trained using the Adam optimizer with a fixed learning rate of `0.05` for `300` epochs. For the unsupervised methods, we use a batch size of `128`. Finally, all bin-based calibration metrics (ECE, MCE, ACE) are computed using a default of 10 bins as specified in our evaluation script. For prompt templates used for evaluation, we present the details in Appendix C.

### A.4  BASELINE DETAILS

For prompt-based baselines, including CAPE (Jiang et al., 2023): a prompt-based method that calibrates next-token probabilities by permuting option order to mitigate LLM biases, Elicitation (Tian et al., 2023): estimates confidence by prompting the model to generate verbalized probabilities. Unsupervised baseline DACA (Luo et al., 2025a) directly aligns the confidence of PoLMs to PLMs on

---

[2] https://huggingface.co/datasets/cais/mmlu
https://huggingface.co/datasets/openlifescienceai/medmcqa
https://huggingface.co/datasets/domenicrosati/TruthfulQA

the agreement samples. Internal Consistency (IC) (Xie et al., 2024b) measures the ratio of consistency between each layer's predictions (mapped to the final vocabulary) and the final layer's output. It is worth noting that the original IC leverages internal consistency within the model's reasoning process. Here, we ignore reasoning and directly generate the final answer for calculation. Since Elicitation and IC can only output confidence for prediction classes, we do not calculate the Brier Score.

# B EVALUATION ON OTHER DOMAINS

In our main experiments, we conduct our evaluation on MMLU (Hendrycks et al., 2021) dataset. To further validate the generalizability of our method, we also present results on the MedMCQA (Pal et al., 2022) dataset, which is from the medical domain. All experimental settings are kept consistent with our main evaluation to ensure a fair comparison. The comprehensive results are shown in Table 6.

| Models | Methods | Evaluation Metrics | | | |
| --- | --- | --- | --- | --- | --- |
| | | ECE (%) ↓ | MCE (%) ↓ | ACE (%) ↓ | Brier Score ↓ |
| LLama3.1-8B | Vanilla | $16.919_{\pm0.699}$ | $27.511_{\pm0.424}$ | $15.679_{\pm1.388}$ | $0.564_{\pm0.005}$ |
| | DACA | $5.149_{\pm0.350}$ | $10.582_{\pm0.521}$ | $5.729_{\pm0.374}$ | $0.517_{\pm0.003}$ |
| | **DUAL-ALIGN (Ours)** | $\mathbf{4.684_{\pm0.171}}$ | $\mathbf{8.881_{\pm0.393}}$ | $\mathbf{5.106_{\pm0.432}}$ | $\mathbf{0.516_{\pm0.003}}$ |
| | TS[†] (oracle) | $1.587_{\pm0.545}$ | $4.929_{\pm2.491}$ | $1.842_{\pm0.444}$ | $0.513_{\pm0.003}$ |
| Qwen2.5-14B | Vanilla | $26.881_{\pm0.631}$ | $39.386_{\pm0.109}$ | $23.303_{\pm0.471}$ | $0.621_{\pm0.010}$ |
| | DACA | $4.904_{\pm0.433}$ | $9.245_{\pm0.270}$ | $8.361_{\pm0.442}$ | $0.529_{\pm0.005}$ |
| | **DUAL-ALIGN (Ours)** | $\mathbf{3.538_{\pm0.924}}$ | $\mathbf{7.507_{\pm0.866}}$ | $\mathbf{3.483_{\pm0.359}}$ | $\mathbf{0.489_{\pm0.006}}$ |
| | TS[†] (oracle) | $3.628_{\pm0.408}$ | $19.972_{\pm8.798}$ | $7.184_{\pm0.950}$ | $0.498_{\pm0.006}$ |
| Gemma-3-27B | Vanilla | $37.084_{\pm0.058}$ | $49.348_{\pm14.837}$ | $34.293_{\pm4.081}$ | $0.748_{\pm0.001}$ |
| | DACA | $26.872_{\pm0.238}$ | $38.685_{\pm1.628}$ | $24.443_{\pm0.497}$ | $0.628_{\pm0.003}$ |
| | **DUAL-ALIGN (Ours)** | $\mathbf{12.940_{\pm0.176}}$ | $\mathbf{29.034_{\pm0.220}}$ | $\mathbf{14.765_{\pm0.292}}$ | $\mathbf{0.537_{\pm0.001}}$ |
| | TS[†] (oracle) | $6.917_{\pm0.278}$ | $28.561_{\pm0.187}$ | $9.317_{\pm0.297}$ | $0.519_{\pm0.002}$ |

Table 6: Performance comparison across different PoLMs and calibration methods on MedMCQA datasets. Lower values indicate better performance. Best results among unsupervised methods are shown in **bold**. "Vanilla" refers to uncalibrated PoLMs. † indicates calibration methods with access to labels. Values are percentages averaged over 3 runs.

## C  EFFECT OF DIFFERENT PROMPTS

To test our framework's robustness against prompt sensitivity, we evaluated four prompt templates (Figure 7). The results in Table 7 confirm that DUAL-ALIGN consistently outperforms the baselines across all variants, demonstrating its effectiveness is not contingent on specific prompt phrasing and is robust to minor instructional changes.

---

**Prompt Variations for Multiple-Choice Questions**

**Prompt Variant A (used in main experiments)**
Select the correct answer for each of the following questions. Respond with the letter only:
[Question]
A: [Option 1] B: [Option 2] C: [Option 3] D: [Option 4]
Answer:

**Prompt Variant B**
The following are multiple-choice questions. Give ONLY the correct option, no other words or explanation:
[Question]
A: [Option 1] B: [Option 2] C: [Option 3] D: [Option 4]
Answer:

**Prompt Variant C**
For the following multiple choice question, provide just the correct letter:
[Question]
A: [Option 1] B: [Option 2] C: [Option 3] D: [Option 4]
Answer:

**Prompt Variant D**
Directly select the correct answer for the following multiple choice question without any explanations:
[Question]
A: [Option 1] B: [Option 2] C: [Option 3] D: [Option 4]
Answer:

---

Figure 7: Four different prompt instructions for a multiple-choice question task.

| Prompt Type | Methods | Evaluation Metrics | | | |
|---|---|---|---|---|---|
| | | ECE (%) $\downarrow$ | MCE (%) $\downarrow$ | ACE (%) $\downarrow$ | Brier Score $\downarrow$ |
| Prompt A | Vanilla | $10.806_{\pm 0.275}$ | $18.602_{\pm 0.212}$ | $11.809_{\pm 0.652}$ | $0.461_{\pm 0.005}$ |
| | DACA | $7.811_{\pm 0.619}$ | $13.824_{\pm 0.667}$ | $8.064_{\pm 0.544}$ | $0.451_{\pm 0.004}$ |
| | **DUAL-ALIGN (Ours)** | $\mathbf{2.871_{\pm 0.308}}$ | $\mathbf{5.587_{\pm 0.648}}$ | $\mathbf{3.222_{\pm 0.306}}$ | $\mathbf{0.441_{\pm 0.004}}$ |
| | TS[†] (oracle) | $1.526_{\pm 0.450}$ | $4.790_{\pm 1.090}$ | $1.985_{\pm 0.609}$ | $0.441_{\pm 0.004}$ |
| Prompt B | Vanilla | $13.271_{\pm 0.375}$ | $23.224_{\pm 0.708}$ | $13.917_{\pm 0.638}$ | $0.472_{\pm 0.006}$ |
| | DACA | $5.530_{\pm 0.627}$ | $10.027_{\pm 1.251}$ | $6.196_{\pm 0.558}$ | $0.444_{\pm 0.003}$ |
| | **DUAL-ALIGN (Ours)** | $\mathbf{1.441_{\pm 0.127}}$ | $\mathbf{8.835_{\pm 0.301}}$ | $\mathbf{2.278_{\pm 0.225}}$ | $\mathbf{0.439_{\pm 0.004}}$ |
| | TS[†] (oracle) | $1.641_{\pm 0.341}$ | $8.820_{\pm 0.132}$ | $2.488_{\pm 0.424}$ | $0.439_{\pm 0.004}$ |
| Prompt C | Vanilla | $10.183_{\pm 0.254}$ | $18.464_{\pm 1.361}$ | $10.859_{\pm 0.587}$ | $0.456_{\pm 0.005}$ |
| | DACA | $6.435_{\pm 0.710}$ | $11.929_{\pm 0.842}$ | $6.830_{\pm 0.785}$ | $0.444_{\pm 0.004}$ |
| | **DUAL-ALIGN (Ours)** | $\mathbf{3.364_{\pm 0.385}}$ | $\mathbf{6.659_{\pm 0.829}}$ | $\mathbf{3.994_{\pm 0.380}}$ | $\mathbf{0.439_{\pm 0.004}}$ |
| | TS[†] (oracle) | $1.387_{\pm 0.237}$ | $6.954_{\pm 1.340}$ | $2.143_{\pm 0.294}$ | $0.437_{\pm 0.004}$ |
| Prompt D | Vanilla | $11.860_{\pm 0.281}$ | $21.147_{\pm 1.020}$ | $13.414_{\pm 0.451}$ | $0.470_{\pm 0.004}$ |
| | DACA DACA | $5.074_{\pm 0.528}$ | $9.856_{\pm 0.162}$ | $5.729_{\pm 0.632}$ | $0.450_{\pm 0.003}$ |
| | **DUAL-ALIGN (Ours)** | $\mathbf{2.523_{\pm 0.410}}$ | $\mathbf{6.792_{\pm 1.148}}$ | $\mathbf{3.031_{\pm 0.087}}$ | $\mathbf{0.445_{\pm 0.003}}$ |
| | TS[†] (oracle) | $1.915_{\pm 0.084}$ | $5.849_{\pm 3.020}$ | $2.370_{\pm 0.449}$ | $0.445_{\pm 0.003}$ |

Table 7: Effects of different prompt instructions on calibration error using Llama3.1-8B on MMLU dataset.

# D  RELIABILITY DIAGRAM OF DIFFERENT BASELINES

This section provides reliability diagrams to visually assess calibration performance across our experiments. These plots show model accuracy versus confidence, with perfect calibration represented by the diagonal line. The following figures (Figure 8 to Figure 13) present these diagrams for the uncalibrated (Vanilla) model, the DACA baseline, our DUAL-ALIGN framework, and the supervised Temperature Scaling (TS) oracle. These visualizations visually confirm the quantitative results from the main paper, clearly illustrating that DUAL-ALIGN significantly reduces the overconfidence of post-trained models and achieves a calibration profile that closely approaches the supervised oracle.

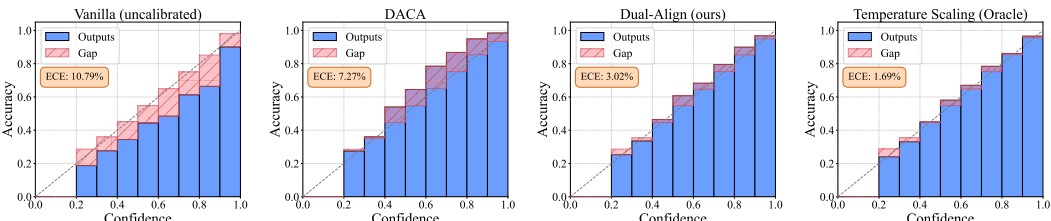

Figure 8: Reliability diagrams of Llama3.1-8B-Instruct on MMLU dataset.

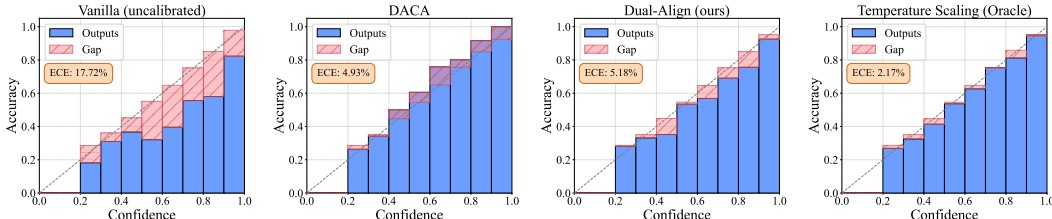

Figure 9: Reliability diagrams of Llama3.1-8B-Instruct on MedMCQA dataset.

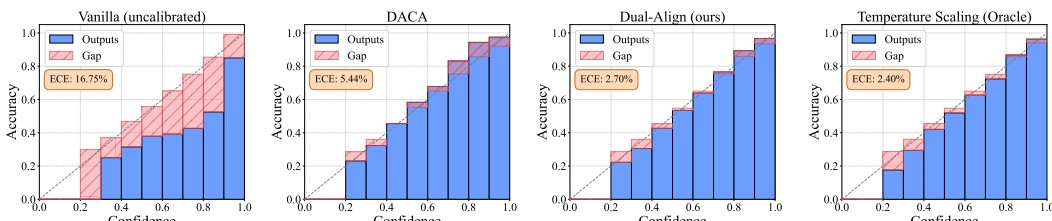

Figure 10: Reliability diagrams of Qwen2.5-14B-Instruct on MMLU dataset.

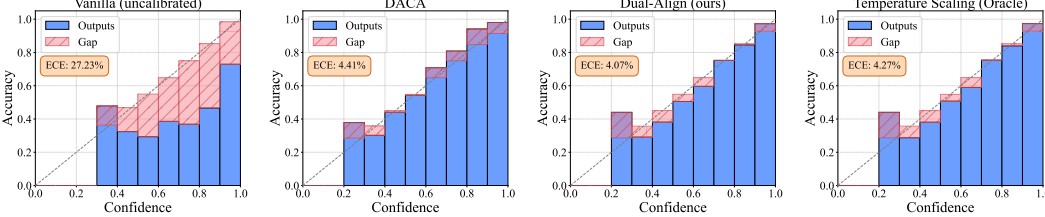

Figure 11: Reliability diagrams of Qwen2.5-14B-Instruct on MedMCQA dataset.

# E  LLM USAGE DISCLOSURE

In accordance with the ICLR 2026 policy on Large Language Model (LLM) usage, we disclose that an LLM (OpenAI GPT-5) was used solely for minor language editing and grammar polishing of the manuscript. The LLM did not contribute to the research ideas, experimental design and data analysis. The authors take full responsibility for the content of this paper.

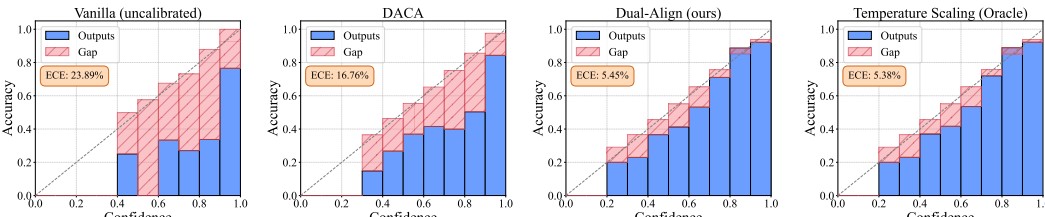

Figure 12: Reliability diagrams of Gemma-3-27b-it on MMLU dataset.

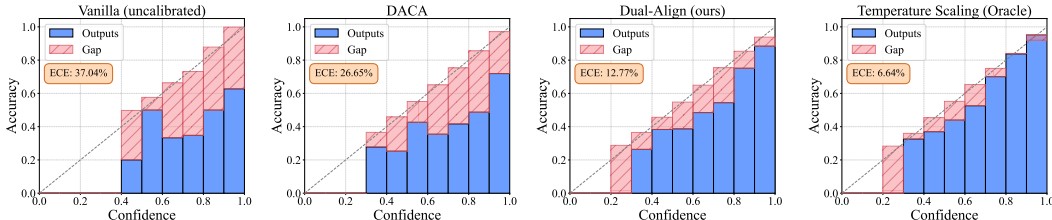

Figure 13: Reliability diagrams of Gemma-3-27b-it on MedMCQA dataset.

