# OpenReview forum: "Unlocking the Pre-Trained Model as a Dual-Alignment Calibrator for Post-Trained LLMs"
_ICLR.cc/2026/Conference — ICLR 2026 Conference Withdrawn Submission_

### Official Review · Reviewer_RrkG · 2025-10-29

**Soundness:** 2
**Presentation:** 3
**Contribution:** 3
**Rating:** 4
**Confidence:** 3

**Summary:**

Beside the widely acknowledged output drift, this paper reveals the existence of “process drift”, which is defined as the confidence discrepancy of pre-trained models (PLMs) and post-trained models (PoLMs) among intermediate layers. The authors recognize such drift as a source of over-confidence and thus causing miscalibration. To address this problem, this work proposes dual-align, a calibration method that learns a temperature parameter that can be applied to mitigate both output drift and process drift. Specifically, this method defines output alignment objective as the KL-Divergence between the temperature-scaled final-layer output distribution of the PoLM and the original distribution of the PLM, and defines process alignment objective as the squared difference between the Inferential Stability Entropy (the entropy computed with intermediate logits). Then the temperature parameter is learned with a combined loss of two objectives. Experiments shows consistent improvement on calibration when applied the learned temperature.

**Strengths:**

1. The revealing of process drift is refreshing.
2. The design of identifying PDL for process alignment is novel but intuitive.
3. Experimental results seem promising in the perspective of calibration.

**Weaknesses:**

- My primary concern is that this work seeks to align the confidence of PLM and PoLM regardless of whether the confidence discrepancy is expected. Post-training itself is designed to alter model behaviors, and the confidence discrepancy between PLM and PoLM is a natural result, which may also indicate that expected behaviors are successfully injected. This work seems to consider this perspective neither theoretically (e.g. discerning what kind of discrepancy should be mitigated) nor empirically (e.g. displaying accuracy/behavior difference of baselines and the proposed method), which makes the contribution of this work questionable.
- Other minor weaknesses lies in the organizing and demonstration of the paper, see Questions.

**Questions:**

1. In section 3 Line 136-137, you are using the final token of input prompt. Does this mean that you expect the first generated token to be the answer token? How is this guaranteed, especially for pre-trained models? Also, this means that no CoT is generated in this process. As far as I know, CoT has an impact on model confidence, and reasoning models are more commonly used recently. Can method proposed in this work used for CoT inference (or it should also be converted into binary classification as stated in section 6)?
2. Figure 5 is a little bit confusing for me. First, what are the meanings of the distribution curves above and aside the main plot? Is it the frequence distribution of ISE/Final Layer Confidence among all samples from some test set? If no, it would be of great help if you can explain further. If yes, high confidence itself does not indicate over-confidence (e.g. in the right figure if samples assigned 0.9 confidence are mostly correct, it could be well-calibrated), making the correlation between low ISE and over-confidence less convincing. Maybe briefly demonstrating relative experimental setting when introducing data plots would make the sections clearer.
3. In Section 4.2 and 4.3, when calculating ISE, is temperature tau only used to divide final layer logits, or used among all layers after the PDL? If it is used among all layers after PDL during training, how it used during inference (applied to final layer or also intermediate layers) ?

---

> ### Author Response · Authors · 2025-11-20
>
> We thank you for recognizing our work as refreshing and novel, intuitive and with promising calibration results. We appreciate the reviewer's comments and suggestions, which we address below:
>
> **[W1] Clarification on aligning the confidence of PLMs and PoLMs.**
>
> We would like to clarify that we are *not* trying to erase all the behavioral differences between PLM and PoLM.
>
> - First, DUAL-ALIGN is a *post-hoc* temperature calibration method: the temperature estimated is only applied to adjust the confidence after the model has completed inference. Therefore, neither the original generation process nor its accuracy / performance is affected. And **any expected behavioral changes introduced by post-training are fully retained.**
> - Our framework does not aim to diminish post-training changes. Instead, it targets only the systematic overconfidence that post-training is empirically known to induce (**Fig.2**). The reviewer is correct that some PLM–PoLM differences are expected and desirable; however, our analysis (**Sec.3, Fig.1–5**) shows that miscalibration arises from two distinct regimes that lead to *several calibration error increase*. Our approach mitigates significantly by learning an additional temperature, not the model decisions (**Tab.1-2**).
>
> We will clarify this distinction more explicitly in the revised version.
>
> **[Q1] Clarification on the next token confidence and Extension to CoT models.**
>
> We thank the reviewer for raising these important clarifications! Yes, we believe that the next token is the model's prediction, using prompts such as:
>
> *Select the correct answer for each of the following questions. Respond with the letter only:[Question]
> A: [Option 1] B: [Option 2] C: [Option 3] D: [Option 4]
> Answer:*
>
> As a result, the model should present the choices after *Answer:*. It is important to clarify that we do not need to assume the pre-trained model to generate the choices; instead, we follow the standard formulation of multiple-choice evaluation, and simply obtain the logits for each option (A, B, C, D) and calculate their softmax probabilities. Previous studies [1,2] have demonstrated that the confidence scores of PLMs obtained in this manner are well-calibrated.
>
> As for CoT models, we believe it is still possible to caluating the confidence by conditioning the choices on the concatenation of input prompt and trace. In addition, we can regard the reasoning and answers generated by these models as answers to open-ended tasks, so we can directly use the p(true) method as Section 6 to obtain and calibrate confidence.
>
> **[Q2] Clarification on Figure 5.**
>
> The curves above and beside the scatter plot are indeed the marginal distributions, i.e., the frequency distribution of ISE/Final Layer Confidence for all MMLU samples on Llama-3.1-8B-Instruct. And the key observation in Fig.5 is not simply the presence of high-confidence points, but the collapse of the PoLM distribution into a narrow cluster of extreme confidences paired with abnormally low ISE values. As shown in **Fig.2 of our submission**, PLM is relatively well-calibrated. In contrast, comparing the confidence distributions of the PoLM and PLM in Figure 5 clearly reveals a significant overconfidence issue of the PoLM.
>
> In other words, the PoLM assigns very high confidence (≈1.0) to many samples where the PLM expresses a healthy range of uncertainty. This can indicate classic overconfidence rather than appropriate high-confidence predictions. Thank you for your suggestion; we will include the experimental settings for each plot in the final version.
>
> **[Q3] Clarification on the use position of temperature.**
>
> Thank you for raising this important nuance. To clarify: First, during training we apply τ to the logits of all intermediate layers after the identified Peak Divergence Layer (PDL). Then, the ISE measure in our approach is solely used as a training‐time signal to detect inference‐trajectory divergence between the PLM and PoLM (after the PDL).
>
> This is because DUAL-ALIGN is a post-hoc calibration method, and our objective is not to alter or intervene in the model’s intermediate‐layer behavior, but rather to leverage ISE to guide the fitting of the temperature parameter τ. At inference time, the model’s internal activations remain unchanged and only the final‐layer logits are scaled by τ for confidence calibration. We will revise the manuscript to more explicitly highlight that “ISE does not impose a change on model inference behavior” to avoid any ambiguity.
>
> [1] OpenAI. Gpt-4 technical report. arXiv preprint arXiv:2303.08774, 2023.
>
> [2] Luo B, et al. Your Pre-trained LLM is Secretly an Unsupervised Confidence Calibrator. NeurIPS 2025.

---

> > ### Author Response · Authors · 2025-11-26
> > **any comments on our response?**
> >
> > Dear reviewer RrkG,
> >
> > We wanted to touch base with you for the author-reviewer discussion. We trust you've had the opportunity to review our rebuttal, and we're eager to address any questions or comments you may have.
> >
> > Thank you once again for your time and dedication to this review process. We look forward to your response and to furthering the dialogue on our manuscript.
> >
> > Thanks,
> >
> > Authors

---

> > > ### Comment · Reviewer_RrkG · 2025-11-26
> > >
> > > Thanks for the further demonstration of authors, which address most of my questions. Now I have only one concern, how the calibrating method affect model performance ? If relavent experimental results are provided and can prove that the method does not severly sacrifice performance, I will consider rasing my score to 6.

---

> > > > ### Author Response · Authors · 2025-11-26
> > > >
> > > > Thank you for your reply! We are very glad to hear that our response has addressed most of your concerns. As for the impact of our method on model performance, we provide the following explanation.
> > > >
> > > > First, it is important to clarify that our method follows the classic temperature scaling [1] to calibrate post-trained language models, without impacting the model’s accuracy. To be more specific, temperature scaling, the simplest extension of Platt scaling, uses a single scalar parameter $T>0$ for all classes. Given the logit vector $\mathbf{z}_i$, the new confidence prediction is
> > > >
> > > > $$
> > > > \hat{q}_i = \max_k \sigma(\mathbf{z}_i/T)^{(k)},
> > > > $$
> > > >
> > > > where $\sigma(\cdot)$ is the softmax function. Because the parameter $T$ does not change the maximum of the softmax function, the class prediction $y_i$ remains unchanged. In other words, temperature scaling does not affect the model’s accuracy.
> > > >
> > > > Furthermore, we conducted experiments on MMLU to verify this.
> > > >
> > > > | Model        | Accuracy% (Ours)| Accuracy% (vanilla) |
> > > > | :------------------  |  :----------- |  :----------- |
> > > > |   Llama3.1-8B    |   65.44  | 65.44  |
> > > > |   Qwen2.5-14B    |   76.19 | 76.19  |
> > > > |   Gemma3-27B     |  75.14    | 75.14 |
> > > >
> > > >
> > > > [1] Guo, Chuan, et al. "On calibration of modern neural networks." International conference on machine learning. PMLR, 2017.

---

> > > > > ### Author Response · Authors · 2025-11-28
> > > > > **Any additional comments**
> > > > >
> > > > > Dear Reviewer RrkG,
> > > > >
> > > > > As suggested, we have provided additional clarification and experimental results in our last response. We are wondering if you have additional thoughts or comments on our response, and we will be happy to further discuss and clarify if needed.
> > > > >
> > > > > Best,
> > > > >
> > > > > Authors

---

> > > > > ### Comment · Reviewer_RrkG · 2025-11-28
> > > > >
> > > > > Thanks for clarifying. I missed the fact that the experiments do not involve sampling (which may affect model behavior when temperature changes).
> > > > >
> > > > > I am willing to raise my rating to 6. However, it seems that OpenReview has banned reviewers from editing the initial rating. I will modify as soon as possible when the edit function reopen.

---

> > > > > > ### Author Response · Authors · 2025-11-28
> > > > > > **Thank you for your response**
> > > > > >
> > > > > > Thank you so much for your quick reply and for increasing the rating to 6! We look forward to that.
> > > > > >
> > > > > > Best,
> > > > > >
> > > > > > Authors

---

### Official Review · Reviewer_9k8u · 2025-10-29

**Soundness:** 3
**Presentation:** 3
**Contribution:** 2
**Rating:** 4
**Confidence:** 4

**Summary:**

This paper proposes DUAL-ALIGN, an unsupervised output calibration method designed to address the previously overlooked issue of process drift. The authors’ motivation is that aligning PoLM and PLM only on consistent responses is insufficient for calibrating inconsistent cases where the underlying reasoning paths bifurcate. To this end, they introduce an unsupervised alignment approach that jointly accounts for output and process drift by minimizing (i) the KL divergence between the output distributions of the PLM and PoLM and (ii) the squared ISE discrepancy accumulated along the reasoning process. The method learns a single scalar temperature that is applied at inference time to calibrate the outputs. Empirically, DUAL-ALIGN outperforms baselines across multiple model scales and families, achieving performance close to supervised methods.

**Strengths:**

1. Clear motivation: The authors identify and thoroughly analyze the process drift overlooked by prior work.
2. Methodological novelty: They propose a new unsupervised algorithm that jointly addresses output drift and process drift, achieving performance that approaches supervised methods.
3. Comprehensive experiments: Evaluations span multiple model scales and families, as well as diverse post-training paradigms.

**Weaknesses:**

When fitting the temperature, the process-drift loss incorporates intermediate-layer logits; however, at inference time only the final layer (i.e., the output) is calibrated. This train–inference mismatch makes the source of the reported gains puzzling. In particular, because inference cannot account for process drift, the learned parameter does not actually stabilize the intermediate layers where such drift occurs, leaving the concrete mechanism behind the performance improvement unclear.

**Questions:**

1. Section 5.3 ablations (Table 3): Table 3 shows that Output Only actually degrades performance. I’m curious whether this is because instances with answer inconsistency (i.e., process drift) were not filtered out—this leads to another question (see Q2).
2. On loss weighting: For samples exhibiting output drift, weighting seems reasonable. However, for process-drift samples (i.e., cases with discrepant final answers), would assigning a weight to Loss output harm performance? Furthermore, if we forgo weighting altogether and simply stratify samples by the two drift types, how would performance change? I would like to see ablations specifically on the weighting strategy.
3. Section 6 (open-ended setting): When the PLM and PoLM produce different answers, their self-evaluation contexts differ. In that case, is cross-model, layer-wise JSD still a robust indicator of “process divergence”? Alternatively, do you still compute it using the last token of the same prompt? This raises another concern: confidence can change during generation, so the prompt’s final token may no longer reflect the confidence at the time the decision is made. Could the authors clarify this? I am particularly interested in the open-ended scenario because it is the most common in practical tasks.

---

> ### Author Response · Authors · 2025-11-20
>
> We are glad to see that the reviewer recognized the strengths of our work from various perspectives. We thank the reviewer for the thorough comments and suggestions. We are happy to clarify as follows:
>
> **[W1] The train-inference mismatch of temperature.**
>
> Thank you for pointing this out! We want to clarify that although the process-drift loss uses intermediate-layer information during training, our method does not aim to calibrate or modify intermediate layers at inference. Instead, our key idea is to utilize the intermediate signals so that the single global temperatire  τ can be better estimated, to reflect how much the post-trained model’s final confidence has been distorted by unstable inference trajectories. Although only the logits are calibrated during inference, its value *is optimized under constraints informed by process drift*, which is why it effectively corrects the downstream overconfidence that originates from internal instability. Our ablations (**Tab.3**) confirm this: output-only alignment is suboptimal, while process-only alignment substantially improves calibration—showing that training-time access to intermediate signals yields a better τ, which then calibrates final outputs without requiring intermediate-layer access at test time.
>
> **[Q1] The performance degradation of the Output Only loss in Table 3.**
>
> We agree with the reviewer that the performance degradation is indeed due to the fact that this objective is applied uniformly to all samples—including those exhibiting process drift, where the PLM and PoLM disagree on the final prediction. For these disagreement cases, aligning the final-layer distributions forces the PoLM to match a conclusion derived from a fundamentally different inference pathway, which empirically produces underconfident or unstable behaviors, as discovered in DACA baseline [1]. This confirms our diagnosis that output drift and process drift cannot be treated uniformly, and explains why Output-Only underperforms DACA even though it optimizes a stronger alignment objective.
>
> **[Q2] Weight ablation for two drift samples.**
>
> Thank you for raising this concern. We conducted additional ablation experiments on MMLU using two alternative weighting strategies. The first strategy involves simply stratifying samples by the two drift types. The second strategy is as follows: for instances exhibiting output drift, we retain the original $L_{\text{Dual-Align}}$, which combines $L_{\text{Output}}$ and $L_{\text{Process}}$ with weighted contributions; for instances with process drift, we optimize only $L_{\text{Process}}$. The average results across three models show that neither of these strategies achieves optimal calibration performance, while our proposed weighting strategy yields the best calibration performance.
>
> | Model        | ECE% (ours)| ECE% (simple stratify) |ECE% (only output weighted) | ECE% (Vanilla) |
> | :------------------  |  :----------- |  :----------- | :----------- | :----------- |
> |   Llama3.1-8B    |   **2.871**   |  5.547 |   5.291     |  10.806      |
> |   Qwen2.5-14B    |   2.423  |  2.580 |     **2.412** |   16.735      |
> |   Gemma3-27B     |   **5.247**   | 17.299 |     17.362   |  23.842       |
> |   Average     |   **3.514**   |  8.475   |   8.355     |  17.128 |
>
> **[Q3] Clarification on open-ended setting.**
>
> We would like to clarify that, in an open-ended setting, we follow prior work [2] use the answers generated by PoLM as context and pass them to PLM and PoLM to judge correctness. Therefore, in fact, PLM does not answer open-ended questions but is only used to provide confidence in judging whether the answer is correct or not. In this case, we can still use next-token probability to represent the model's confidence. However, we do believe the reviewer's suggestion is useful and can be a nice extension of our work in the future!
>
> [1] Luo B, et al. Your Pre-trained LLM is Secretly an Unsupervised Confidence Calibrator. NeurIPS 2025.
>
> [2] Anthropic, Language Models (Mostly) Know What They Know, 2022

---

> > ### Author Response · Authors · 2025-11-26
> > **any comments on our response?**
> >
> > Dear reviewer 9k8u,
> >
> > We wanted to touch base with you for the author-reviewer discussion. We trust you've had the opportunity to review our rebuttal, and we're eager to address any questions or comments you may have.
> >
> > Thank you once again for your time and dedication to this review process. We look forward to your response and to furthering the dialogue on our manuscript.
> >
> > Thanks,
> >
> > Authors

---

### Official Review · Reviewer_6TZj · 2025-10-31

**Soundness:** 1
**Presentation:** 2
**Contribution:** 2
**Rating:** 2
**Confidence:** 4

**Summary:**

This paper studies the calibration degradation that occurs when LLMs are post-trained (e.g., via instruction tuning or RLHF). The authors identify two empirical phenomena, i.e., output drift (the model’s confidence inflates while its intermediate reasoning remains consistent with the base model) and process drift (the model’s intermediate activations diverge sharply at a “peak divergence layer”).
Building on this diagnosis, they propose DUAL-ALIGN, an unsupervised post-hoc calibration framework that performs (i) output alignment to correct surface-level overconfidence and (ii) process alignment to restore stability of intermediate inference dynamics. The method learns a single temperature parameter without labels and incurs no extra inference cost, achieving lower calibration errors across several model families (Llama-3.1, Qwen-2.5, Gemma-3) compared with existing unsupervised baselines.

**Strengths:**

- The paper offers an interesting empirical observation by decomposing post-training miscalibration into two distinct regimes (“output drift” and “process drift”). This provides a more granular view of how post-training affects model confidence than prior work, and it may inspire further interpretability-based calibration research.

- The proposed method improves calibration with very low computational cost. It only learns a single scalar temperature parameter while exploiting existing model representations. There is no fine-tuning or additional inference overhead, making it practical for large-scale LLMs.

- The approach is completely label-free. By leveraging the pre-trained model as a self-supervised reference, it eliminates the need for human-annotated validation data and remains effective across several architectures and datasets.

**Weaknesses:**

- The two-drift observation is purely empirical and lacks a principled foundation. The distinction between output and process drift is derived from layer-wise diagnostics rather than foundamental analysis, leaving open questions about whether these patterns generalize to other architectures, training recipes, or datasets. Strengthening this with a more formal analysis or cross-model verification would make the argument more convincing.

- The paper does not evaluate potential side effects on task performance. Although temperature rescaling does not alter argmax predictions, most generation setups rely on top-p sampling, where changing the temperature reshapes token probabilities and can influence output quality. Without reporting accuracy or generation-quality metrics, it remains unclear whether the calibration improvements come at the cost of degraded task performance.

**Questions:**

see above

---

> ### Author Response · Authors · 2025-11-20
>
> We thank the reviewer for the comments and suggestions. We are encouraged that you recognize the core benefits of our approach, which is in-depth diagnosis of post-training on calibration, low computational cost and unsupervised. We address your questions below:
>
>
> **[W1] Cross-model verification for two-drift observation.**
>
> Thank you for raising the concern! First, we clarify that the experiment in Figure 1 is conducted with Llama-3.1-8B on MMLU. During rebuttal, we verify our observation by performing the same experiment on Gemma-3-27B. The table below shows that the two-drift observation can indeed generalize to different models. In the table, Δ means the confidence difference between the pre-trained and post-trained model. The results show that Gemma-3-27B also exhibits a large gap between pre-trained confidence and post-trained confidence in the later layers on disagreement samples, while this phenomenon does not occur on agreement samples. This is consistent with the phenomenon we observed in Figure 1.
>
> Moreover, we want to highlight the main results in **Tab.1, 2 and Fig.6b** have already included evaluations across different models, datasets and post-training recipes, where the consistently better performance than baselines can support our method rationale.
>
> | Layer | Δ Agreement | Δ Disagreement |
> |-------|-------------|----------------|
> | 0     | 0           | 0              |
> | 10    | +0.000006   | -0.000005      |
> | 20    | +0.008989   | -0.019297      |
> | 30    | +0.001331   | -0.002467      |
> | 40    | +0.066876   | +0.362292      |
> | 50    | +0.071054   | +0.310466      |
> | 60    | +0.072620   | +0.400823      |
> | 62    | +0.021250   | +0.910602      |
>
> **[W2] The potential side effects of temperature scaling on task performance.**
>
> Good point raised! We want to kindly remind the reviewer that our approach is for *post-hoc* LLM calibration. The temperature estimated is only applied to adjust the confidence after the model has completed inference. Therefore, the original generation process and its quality and performance is not affected.

---

> > ### Author Response · Authors · 2025-11-26
> > **any comments on our response?**
> >
> > Dear reviewer 6TZj,
> >
> > We wanted to touch base with you for the author-reviewer discussion. We trust you've had the opportunity to review our rebuttal, and we're eager to address any questions or comments you may have.
> >
> > Thank you once again for your time and dedication to this review process. We look forward to your response and to furthering the dialogue on our manuscript.
> >
> > Thanks,
> >
> > Authors

---

### Note · Authors · 2025-12-27

**Comment:**

Dear Reviewers, AC and PC,

We highly appreciate your constructive comments and questions. As we find the key question deserves careful scrutiny and requires further clarification, the authors have decided to withdraw the submission for this round.

Thank you once again for your time and effort in going over our manuscript, not to mention the insightful comments to help us improve the paper.

Best regards,

Authors.

**Withdrawal Confirmation:**

I have read and agree with the venue's withdrawal policy on behalf of myself and my co-authors.